# Interval Methods with Fifth Order of Convergence for Solving Nonlinear Scalar Equations

**Ivan G. Ivanov** [1],*,† and **Tonya Mateva** [2],†

1   Faculty of Economics and Business Administration, Sofia University St. Kliment Ohridski, Sofia 1113, Bulgaria
2   Kolej Dobrich, Konstantin Preslavsky University of Shumen, Shumen 9712, Bulgaria; t.mateva@shu.bg
*   Correspondence: i_ivanov@feb.uni-sofia.bg; Tel.: +35-989-863-6339
†   This work was carried out in collaboration between both authors. Both authors read and approved the final manuscript.

**Abstract:** In this paper, based on Kou's classical iterative methods with fifth-order of convergence, we propose new interval iterative methods for computing a real root of the nonlinear scalar equations. Some numerical experiments have executed with the program INTLAB in order to confirm the theoretical results. The computational results have described and compared with Newton's interval method, Ostrowski's interval method and Ostrowski's modified interval method. We conclude that the proposed interval schemes are effective and they are comparable to the classical interval methods.

**Keywords:** Kou's interval method; Newton's interval method; Ostrowski's interval method; INTLAB; MATLAB

## 1. Introduction

Several iterative methods are used to solve nonlinear equations, the most famous being Newton's method. To improve the order of convergence, many authors have proposed modifications to the Newton formula, obtained by various techniques and with different order of convergence [1–9] and references related to higher-order iterative methods [10–12]. In addition to the classic iteration schemes, interval methods are also used to solve nonlinear equations. Interval analysis is formally presented by Moore [13]. In article [14], the authors have described iterative formulas proposed by Ostrowski [15] and based on Newton's interval method and offer interval modifications of these iterative patterns.

In this article, we discuss some modifications of Newton's method and we present them in an interval form. Kou's classical methods, with fifth order convergence, give very good results, when compared to other classic iterative schemes. We modify Kou's method in interval form to see are they fast and stable in this interval form. We use the methodology for the research, analysis and comparison of iterative schemes applied by authors in articles [14,16] and others. For each of proposed methods, a theorem of convergence and a proof of the theorem is shown. With the proposed methods, we have done experiments with the computer program Matlab and the INTLAB toolbox [17] to evaluate their performance. The second aim is to compare the new interval schemes to Newton's well-known interval scheme and the schemes described in the article [18]. We look at features reviewed in articles [14] and [19] to assess the effectiveness of the offered modifications.

## 2. Background and Methodology

### 2.1. Interval Newton's Method

As a basis for our research we use the methods discussed in the article [14]. We look for a root of a real function $f(x)$. Suppose the first derivative of the function $f'(x)$ is continuous in the range

$[a, b]$ and $f(a).f(b) < 0$. The most famous and preferred method for solving nonlinear equations is Newton's method. Its interval form is: If $x^{(k)}$ is the interval, where the root is located, then we can narrow the formula interval

$$x^{(k+1)} = x^{(k)} \cap N(x^{(k)}), \ k = 0, 1, 2, ... \tag{1}$$

$$N(x) = mid(x) - \frac{f(mid(x))}{f'(x)},$$

where $mid(x) = \frac{a+b}{2}$—the middle of interval $x$. Thus, at each step we get a new, smaller interval and after n in number steps, we reach the root of the function. The standard notations are the following. A real interval is a set of real numbers $x = [\underline{x}, \overline{x}]$, where $\underline{x}$ and $\overline{x}$ are the lower and upper bounds of the interval number $x$, respectively. A real number $x$ is identified with a point interval $x = [\underline{x}, \overline{x}]$. The quality of interval analysis is measured by the width of the interval results, and a sharp enclosure for the exact solution is desirable. The mid-point and the width of an interval $x$ are denoted by $mid(x) = \frac{\underline{x}+\overline{x}}{2}$, and $wid(x) = \underline{x} - \overline{x}$, respectively.

The next two theorems prove the convergence of the method.

**Theorem 1** ([13]). *If an interval $x^{(0)}$ contains a zero $x^*$ of $f(x)$, then so does $x^{(k)}$ for all $k = 0, 1, 2, ...$, defined by (1). Furthermore, the intervals $x^{(k)}$ form a nested sequence converging to $x^*$ if $0 \notin f'(x^{(0)})$.*

**Theorem 2** ([13]). *Given a real rational function f of a single real variable x with rational extensions f, f' of f, f', respectively, such that f has a simple zero $x^*$ in an interval $x^{(0)}$ for which $f(x^{(0)})$ is defined and $f'(x^{(0)})$ is defined and does not contain zero i.e. $0 \notin f'(x^{(0)})$. Then there is a positive real number C such that*

$$wid(x^{(k+1)}) \leq C.(wid(x^{(k)}))^2.$$

If $0 \notin f'(x^{(0)})$, then $0 \notin f'(x^{(k)})$ for all $k$ and mid $(x^{(k)})$ is not contained in $N(x^{(k)})$, unless $f(mid(x^{(k)})) = 0$. So, convergence of the sequence follows [13,20,21]. It should be noted that some special cases of (1) have been discussed in [14] in more details.

Ostrowski offers two classic methods for solving nonlinear equations that the authors of the article [14] modify as interval.

*2.2. Interval Ostrowski's Method*

The solution of the equation $f(x) = 0$, on a given interval $x$ is considered:

$$x^{(k+1)} = x^{(k)} \cap S(x^{(k)}, y^{(k)}), \ k = 0, 1, 2, ..., \tag{2}$$

where

$$N(x) = mid(x) - \frac{f(mid(x))}{f'(x)},$$

$$y^{(k)} = x^{(k)} \cap N(x^{(k)}),$$

$$\lambda = \frac{f(mid(x))}{[f(mid(x)) - 2f(mid(y))].f'(x)},$$

$$S(x, y) = mid(y) - \lambda.f(mid(y)).$$

The following two theorems prove the conditions for the presence of a root in the selected initial interval and the convergence of the method.

**Theorem 3** ([16]). *Assume $f \in C(x^{(0)})$ and $0 \notin f'(x^{(k)})$ for $k = 0, 1, 2, \ldots$ . If $x^{(0)}$ contains a root $x^*$ of $f$, then so do all intervals $x^{(k)}$, $k = 1, 2, \ldots$. Besides, the intervals $x^{(k)}$ form a nested sequence converging to $x^*$.*

**Theorem 4** ([16]). *Suppose $f \in C(x^{(0)})$ and $0 \notin f'(x^{(k)})$ for $k = 0, 1, 2, \ldots$ .*

1.  *If $x^* \in x^{(0)}$ and $S(x^{(k)}, y^{(k)}) \subseteq x^{(k)}$, then $x^{(k)}$ contains exactly one zero of $f$.*
2.  *If $x^{(k)} \cap S(x^{(k)}, y^{(k)}) = \oslash$, then $x^{(k)}$ does not contain any zero of $f$.*

Theorems 3 and 4 are valid for all the methods discussed below. The proof can be seen in [14]. The next theorem shows, that the convergence rate of this method is fourth order.

**Theorem 5** ([14]). *Assume $f \in C(x^{(0)})$ and $0 \notin f'(x^{(k)})$, and $f$ has a unique simple root $x^* \in x^{(0)}$. Then, if $S(x^{(k)}, y^{(k)}) \subseteq x^{(k)}$, the sequence (2) has convergent rate four, i.e., there exists a constant $K$ such that*

$$wid(x^{(k+1)}) \leq K.(wid(x^{(k)}))^4.$$

The proof can be seen in [14].

The next method is a modification of Ostrowski's method with a higher order of convergence. From the experiments in article [14] and in the Section 4 of this article, it is apparent that it is faster than Ostrowski's method and Newton's method.

*2.3. Interval Modified Ostrowski Method*

The interval modified Ostrowski method is introduced by Bakhtiari et al. [14]. We seek a solution of the equation $f(x) = 0$, on interval $x = [\underline{x}, \overline{x}]$.

$$x^{(k+1)} = x^{(k)} \cap M(x^{(k)}, y^{(k)}, z^{(k)}), \ k = 0, 1, 2, \ldots, \tag{3}$$

where

$$N(x) = mid(x) - \frac{f(mid(x))}{f'(x)},$$

$$y^{(k)} = x^{(k)} \cap N(x^{(k)}),$$

$$\lambda = \frac{f(mid(x))}{[f(mid(x)) - 2f(mid(y))].f'(x)},$$

$$S(x, y) = mid(y) - \lambda.f(mid(y)),$$

$$z^{(k)} = x^{(k)} \cap S(x^{(k)}, y^{(k)}),$$

$$M(x, y, z) = mid(z) - \lambda.f(mid(z)).$$

The next theorem shows, that the convergence rate of this method is of sixth order.

**Theorem 6** ([14]). *Assume $f \in C(x^{(0)})$ and $0 \notin f'(x^{(k)})$, and $f$ has a unique simple root $x^* \in x^{(0)}$. Then, if $M(x^{(k)}, y^{(k)}, z^{(k)}) \subseteq x^{(k)}$, the sequence (3) has convergent rate of six, i.e., there exists a constant $K$ such that*

$$wid(x^{(k+1)}) \leq K(wid(x^{(k)}))^6.$$

The proof can be seen in [14].

## 3. Main Results

In this section three classical interval methods for finding a simple zero of a nonlinear equation are modified. Convergence rates of the proposed methods are derived. We look at three iterative schemes that we modify in the interval form and with them we explore selected functions to find the number of iterations needed to find their roots at a pre-selected interval. Moreover, we select the initial approximations $x_n$ following the classical approach, i.e., which satisfy the following conditions:

1. We explore the functions in the interval $[a, b]$, for which $f(a) \cdot f(b) < 0$, which ensure that there is a root in the corresponding interval.

2. We require in the corresponding interval, the first and the second derivative to be continuous for $\forall x \in [a, b]$: $f'(x) \neq 0$ and $f''(x) \neq 0$.

3. For initial approximation we use this end of the interval for which

$$f(a) \cdot f''(a) > 0.$$

In the interval form, the initial approximation $x^{(k)}$ is interval, in which $f$ has a simple root $x^*$ and $f'(x^{(k)})$ is defined and does not contain zero i.e., $0 \notin f'(x^{(k)})$.

### 3.1. First Method of Kou in Classic and Interval Form

The iterative formula offered by Kou in [18] is a modification of Newton's formula. The classical Kou's method given by the formula:

$$x_{n+1} = u_{n+1} - \frac{f(u_{n+1})}{f'(x_{n+1}^*)}, \tag{4}$$

where

$$u_{n+1} = x_n - 2.\frac{f(x_n)}{f'(x_n) + f'(x_{n+1}^*)}, \qquad x_{n+1}^* = x_n - \frac{f(x_n)}{f\prime(x_n)}.$$

A detailed description of this method is given in [18]. The efficiency index of Kou method (4) is 1.495. We modify Formula (4) in the interval form as follows. We seek a solution of the equation $f(x) = 0$, on interval $x = [\underline{x}, \overline{x}]$. Interval extension is introduced:

$$x^{(k+1)} = x^{(k)} \cap S(x^{(k)}, y^{(k)}, z^{(k)}), \ k = 0, 1, 2, ..., \tag{5}$$

where

$$y^{(k)} = x^{(k)} \cap N(x^{(k)}),$$

$$N(x) = mid(x) - \frac{f(mid(x))}{f'(x)},$$

$$z^{(k)} = x^{(k)} \cap M(x^{(k)}, y^{(k)}),$$

$$M(x, y) = mid(x) - \frac{2.f(mid(x))}{f'(x) + f'(y)},$$

$$S(x, y, z) = mid(z) - \lambda.f(mid(z)), \qquad \lambda = \frac{1}{f'(y)}. \tag{6}$$

Under some simple conditions, it is possible to derive that the convergence rate of the introduced interval method (5) is fifth order in the next theorem.

**Theorem 7.** *Assume $f \in C(x^{(0)})$ and $0 \notin f'(x^{(k)})$, and $f$ has a unique simple root $x^* \in x^{(0)}$. Then, if $S(x^{(k)}, y^{(k)}, z^{(k)}) \subseteq x^{(k)}$, the sequence (5) has convergent rate of fifth order, i.e., there exists a constant $K$ such that*

$$wid(x^{(k+1)}) \leq K.(wid(x^{(k)}))^5.$$

**Proof.** By Mean Value Theorem we have

$$f(mid(x^{(k)})) = f'(\xi) \left[ mid(x^{(k)}) - x^* \right],$$

where $\xi$ is between $mid(x^{(k)})$ and $x^*$. Since $S(x^{(k)}, y^{(k)}) \subset x^{(k)}$, thus from (5) and (6) we have

$$x^{(k+1)} = mid(z^{(k)}) - \lambda \left[ mid(z^{(k)}) - x^* \right] . f'(\xi), \tag{7}$$

where

$$\lambda = \frac{1}{f'(y^{(k)})}. \tag{8}$$

From (7) we have

$$wid(x^{(k+1)}) = wid(\lambda). \left| mid(z^{(k)}) - x^* \right| . \left| f'(\xi) \right| . \tag{9}$$

It is clear that

$$\left| mid(z^{(k)}) - x^* \right| \leq wid(z^{(k)}).$$

Since $z^{(k)} = x^{(k)} \cap M(x^{(k)}, y^{(k)})$, and $M(x, y)$ is the iterative method of Weerakoon and Fernando with third order convergence [9], there are positive constants $K_i$, $i = 1, 2...$ such that

$$wid(z^{(k)}) \leq K_1(wid(x^{(k)}))^3. \tag{10}$$

Moreover (8) gives

$$wid(\lambda) = wid \left( \frac{1}{f'(y^{(k)})} \right) \leq wid(y^{(k)}) \leq K_2.(wid(x^{(k)}))^2. \tag{11}$$

Now, let $|f'(\xi)| \leq K_3$ and using the relation (10), (11) in (9), we have the following error bounds as

$$wid(x^{(k+1)}) \leq K(wid(x^{(k)}))^5.$$

The proof is completed.  □

### 3.2. Second Method of Kou in Classic and Interval Form

The iterative formula offered by Kou in [18] is a modification of the Newton's formula. The classical Kou's method given by the formula:

$$x_{n+1} = u_{n+1} - \frac{f(u_{n+1})}{2.f'(\frac{1}{2}.(x_n + x_{n+1}^*)) - f'(x_n)}, \tag{12}$$

where

$$u_{n+1} = x_n - \frac{f(x_n)}{f'(\frac{1}{2}.(x_n + x_{n+1}^*))}, \qquad x_{n+1}^* = x_n - \frac{f(x_n)}{f'(x_n)}$$

with the efficiency index 1.495.

We propose the following modification in the interval form for (12). We seek a solution of the equation $f(x) = 0$, on the interval $x = [\underline{x}, \overline{x}]$. We chose the initial interval $x^{(0)}$ and assuming that we know interval $x^{(k)}$ we compute next approximation using the iterative formula

$$x^{(k+1)} = x^{(k)} \cap S(x^{(k)}, y^{(k)}, z^{(k)}), \quad k = 0, 1, 2, ..., \tag{13}$$

where

$$N(x) = mid(x) - \frac{f(mid(x))}{2.f'(x)},$$

$$y^{(k)} = x^{(k)} \cap N(x^{(k)}),$$

$$M(x,y) = mid(x) - \frac{f(mid(x))}{f'(y)},$$

$$z^{(k)} = x^{(k)} \cap M(x^{(k)}, y^{(k)}),$$

$$S(x,y,z) = mid(z) - \lambda.f(mid(z)), \quad \lambda = \frac{1}{2.f'(y) + f'(x)}. \tag{14}$$

The next theorem proves the convergence of the method:

**Theorem 8.** *Assume* $f \in C(x^{(0)})$ *and* $0 \notin f'(x^{(k)})$, *and* $f$ *has a unique simple root* $x^* \in x^{(0)}$. *Then, if* $S(x^{(k)}, y^{(k)}, z^{(k)}) \subseteq x^{(k)}$, *the sequence* (13) *has convergent rate of fifth order, i.e., there exists a constant* $K$ *such that*

$$wid(x^{(k+1)}) \leq K.(wid(x^{(k)}))^5.$$

**Proof.** By the Mean Value Theorem we have

$$f(mid(x^{(k)})) = f'(\xi) \left[ mid(x^{(k)}) - x^* \right],$$

where $\xi$ is between $mid(x^{(k)})$ and $x^*$. Since $S(x^{(k)}, y^{(k)}) \subset x^{(k)}$, thus from (13) and (14) we have

$$x^{(k+1)} = mid(z^{(k)}) - \lambda \left[ mid(z^{(k)}) - x^* \right].f'(\xi), \tag{15}$$

where

$$\lambda = \frac{1}{2.f'(y^{(k)}) + f'(x)}. \tag{16}$$

From (15) we have

$$wid(x^{(k+1)}) = wid(\lambda). \left| mid(z^{(k)}) - x^* \right| . \left| f'(\xi) \right|. \tag{17}$$

It is clear that

$$\left| mid(z^{(k)}) - x^* \right| \leq wid(z^{(k)}).$$

Since $z^{(k)} = x^{(k)} \cap M(x^{(k)}, y^{(k)})$, and $M(x,y)$ is the iterative scheme of Frontini and Sormani with third order convergence [3], there are positive constants $K_i$, $i = 1, 2...$ such that

$$wid(z^{(k)}) \leq K_1(wid(x^{(k)}))^3. \tag{18}$$

Moreover (16) gives

$$\begin{aligned} wid(\lambda) &= wid\left(\frac{1}{2.f'(y^{(k)}) + f'(x^{(k)})}\right) \\ &\leq wid(x^{(k)} + y^{(k)}) \leq \left(wid(x^{(k-1)})\right)^2 + \left(wid(x^{(k)})\right)^2. \end{aligned} \tag{19}$$

Now, let $|f\prime(\xi)| \leq K_3$ and using the relation (18), (19) in (17), we have the following error bounds as

$$wid(x^{(k+1)}) \leq K. \left[ \left(wid(x^{(k-1)})\right)^2 + \left(wid(x^{(k)})\right)^2 \right] .wid(x^{(k)})^3.$$

**Remark 1.** *By our reasoning, we assume, that*

$$wid(x^{(k)} + y^{(k)}) \leq \left(wid(x^{(k-1)})\right)^2 + \left(wid(x^{(k)})\right)^2 \leq \left(wid(x^{(k)})\right)^2,$$

*and the local order of convergence of the interval Kou's method is fifth order, but we have no theoretical proof of that.*

□

### 3.3. Third Method of Kou in Classic and Interval Form

The iterative formula offered by Kou in [18] is a modification of Newton's formula. The classical form of Kou's method is:

$$x_{n+1} = u_{n+1} - \frac{f(u_{n+1})}{f'(x^*_{n+1})} \quad \text{with} \tag{20}$$

$$u_{n+1} = x_n - \frac{f(x_n)}{2} \cdot \left(\frac{1}{f\prime(x_n)} + \frac{1}{f'(x^*_{n+1})}\right), \qquad x^*_{n+1} = x_n - \frac{f(x_n)}{f\prime(x_n)}.$$

We modify Formula (20) as interval form. We seek a solution of the equation $f(x) = 0$, on interval $x = [\underline{x}, \overline{x}]$. Interval extension of the classic form of Kou's method is introduced as

$$x^{(k+1)} = x^{(k)} \cap S(x^{(k)}, y^{(k)}, z^{(k)}), \; k = 0, 1, 2, ..., \tag{21}$$

where

$$N(x) = mid(x) - \frac{f(mid(x))}{f'(x)},$$

$$y^{(k)} = x^{(k)} \cap N(x^{(k)}),$$

$$M(x, y) = mid(x) - \frac{f(mid(x))}{2} \cdot \left(\frac{1}{f'(x)} + \frac{1}{f'(y)}\right),$$

$$z^{(k)} = x^{(k)} \cap M(x^{(k)}, y^{(k)}),$$

$$S(x, y, z) = mid(z) - \lambda.f(mid(z)) \tag{22}$$

$$\lambda = \frac{1}{f'(y)}.$$

The next theorem proves the convergence of the method:

**Theorem 9.** *Assume $f \in C(x^{(0)})$ and $0 \notin f'(x^{(k)})$, and $f$ has a unique simple root $x^* \in x^{(0)}$. Then, if $S(x^{(k)}, y^{(k)}, z^{(k)}) \subseteq x^{(k)}$, the sequence (21) has convergent rate of fifth order, i.e., there exists a constant K such that*

$$wid(x^{(k+1)}) \leq K.(wid(x^{(k)}))^5.$$

**Proof.** By Mean Value Theorem we have

$$f(mid(x^{(k)})) = f'(\xi)\left[mid(x^{(k)}) - x^*\right],$$

where $\xi$ is between $mid(x^{(k)})$ and $x^*$. Since $S(x^{(k)}, y^{(k)}) \subset x^{(k)}$, thus from (21) and (22) we have

$$x^{(k+1)} = mid(z^{(k)}) - \lambda \left[mid(z^{(k)}) - x^*\right].f'(\xi), \tag{23}$$

with

$$\lambda = \frac{1}{f'(y^{(k)})}. \tag{24}$$

According to (23) we receive

$$wid(x^{(k+1)}) = wid(\lambda).\left|mid(z^{(k)}) - x^*\right|.\left|f'(\xi)\right|. \tag{25}$$

It is clear that

$$\left|mid(z^{(k)}) - x^*\right| \leq wid(z^{(k)}).$$

Since $z^{(k)} = x^{(k)} \cap M(x^{(k)}, y^{(k)})$, and $M(x,y)$ is the iterative scheme of Homeier with third order convergence [5], there are positive constants $K_i$, $i = 1, 2...$ such that

$$wid(z^{(k)}) \leq K_1(wid(x^{(k)}))^3. \tag{26}$$

Moreover (24) gives

$$wid(\lambda) = wid\left(\frac{1}{f'(y^{(k)})}\right) \leq wid(y^{(k)}) \leq K_2.(wid(x^{(k)}))^2. \tag{27}$$

Now, let $\left|f'(\xi)\right| \leq K_3$ and using the relation (26), (27) in (25), we have the following error bounds as

$$wid(x^{(k+1)}) \leq K(wid(x^{(k)}))^5.$$

□

## 4. Numerical Experiments

All calculations were made using a computer program MATLAB7.6.0(R2008a) and INTLAB toolbox—Application for interval analysis, created by Rump [17]. The desired accuracy is $tol = 1.0 \times 10^{-15}$. We select initial interval $X = [a, b]$, which satisfies the following conditions:

1. $f(a).f(b) < 0$ and 2. $f'(x)$ is continuous in the range $[a, b]$.

We use the stop criterion $wid(x^{(k)}) < tol$ for all interval methods. The results are given in Table 1. The column I contains the function, the interval in which we examine it, and the root in that interval. Column II contains the methods, which we compare. Columns III and V contain the intervals, in which we examine the function, and columns IV and VI contain the corresponding number of iterations. We introduce academic examples in functions $f_1, f_2, \ldots, f_8$ and second, we apply the introduced interval iterative methods to Van der Waals' equation [22] presented through $f_9, f_{10}$ functions.

Article [14] presents Ostrowski's interval method, Ostrowski's interval modified method, and Newton's interval method. The conclusions of the experiments with these three methods show that Ostrowski's modified interval method is the fastest. We see that Kou's three methods are just as fast and effective for the examples considered in the experiments. For some functions, the interval method Kou2 is divergent, however Kou1 and Kou3 make a small number of iterations for all considered functions. For some features, these two methods have even a smaller number of iteration than modified interval Ostrowski's method. We can summarize that Kou1 and Kou3 methods are faster than Newton's base method, Ostrowski's method, Ostrowski's modified method for all the functions at both selected intervals. We can say that by comparing these two methods, they are the fastest approaching and are stable for any choice of different intervals and functions.

**Table 1.** Experiments with some functions.

| I | II | III | IV | V | VI |
|---|---|---|---|---|---|
| $f_1(x) = x(x^9 - 1) - 1$ | Newton | $[1, 1.5]$ | 6 | $[0.8, 5.5]$ | 10 |
| Root: $x_n = 1.0757661$ | Ostrowski | | 4 | | Nan |
| Space containing the root: | Ostr– modif | | 3 | | 6 |
| $(0.8, \infty)$ | Kou1 | | 3 | | 5 |
| | Kou2 | | Nan | | - |
| | Kou3 | | 3 | | 6 |
| $f_2(x) = x^2 - \exp(x) - 3x + 2$ [18] | Newton | $[0, 1]$ | 5 | $[-1, 1.5]$ | 4 |
| Root: $x_n = 0.257530$ | Ostrowski | | 3 | | 3 |
| Space containing the root: | Ostr– modif | | 2 | | 3 |
| $(-\infty, +\infty)$ | Kou1 | | 2 | | 2 |
| | Kou2 | | Nan | | 1 |
| | Kou3 | | 2 | | 2 |
| $f_3(x) = \exp(-x) + \cos x$ | Newton | $[1, 2]$ | 4 | $[0.5, 2.5]$ | 5 |
| Root: $x_n = 1.7461395$ | Ostrowski | | 3 | | 3 |
| Space containing the root: | Ostr- modif | | 2 | | 3 |
| $(-\infty, 3.1)$ | Kou1 | | 2 | | 3 |
| | Kou2 | | 2 | | 3 |
| | Kou3 | | 2 | | 3 |
| $f_4 = \exp x - 4x^2$ | Newton | $[4, 5]$ | 7 | $[4, 6]$ | 8 |
| Root: $x_n = 4.306584$ | Ostrowski | | Nan | | Nan |
| Space containing the root: | Ostr– modif | | Nan | | Nan |
| $(3.3, +\infty)$ | Kou1 | | 4 | | 4 |
| | Kou2 | | Nan | | Nan |
| | Kou3 | | 4 | | 4 |
| $f_5(x) = (x + 2). \exp x - 1$ | Newton | $[-1, 0]$ | 5 | $[-2, 5]$ | 7 |
| Root: $x_n = -0.442854$ | Ostrowski | | 3 | | Nan |
| Space containing the root: | Ostr– modif | | Nan | | 5 |
| $(-2.8, +\infty)$ | Kou1 | | 2 | | 4 |
| | Kou2 | | Nan | | - |
| | Kou3 | | 2 | | 3 |
| $f_6(x) = \cos x - x$ [18] | Newton | $[0, 1]$ | 4 | $[-1, 2]$ | 5 |
| Root: $x_n = 0.739085$ | Ostrowski | | 3 | | 4 |
| Space containing the root: | Ostr- modif | | 1 | | 3 |
| $(-1.5, 4.6)$ | Kou1 | | 2 | | 3 |
| | Kou2 | | 3 | | 3 |
| | Kou3 | | 2 | | 3 |
| $f_7(x) = \frac{2}{x^5} + 3 * sin(x^4) + 5$ | Newton | $[-1, -0.5]$ | 6 | $[-1, -0.1]$ | 8 |
| Root: $x_n = -0.79793001477798$ | Ostrowski | | 5 | | 4 |
| Space containing the root: | Ostr- modif | | 5 | | 5 |
| $(-1, +\infty)$ | Kou1 | | 4 | | 4 |
| | Kou2 | | Nan | | - |
| | Kou3 | | 4 | | 4 |
| $f_8(x) = (x - 2)^{23} - 1$ | Newton | $[2.7, 4]$ | 7 | $[2.7, 5]$ | 10 |
| Root: $x_n = 3.00000000000000$ | Ostrowski | | Nan | | Nan |
| Space containing the root: | Ostr- modif | | Nan | | 5 |
| $(2.7, +\infty)$ | Kou1 | | 4 | | 4 |
| | Kou2 | | Nan | | Nan |
| | Kou3 | | 4 | | 6 |
| $f_9(V) = 10\,V^3 - 24.64917\,V^2$ $+1.36\,V - 0.00432888$ [22] | Newton | $[2.2, 2.9]$ | 7 | $[2.2, 2.6]$ | 5 |
| | Ostrowski | | Nan | | Nan |
| Root: $V_n = 2.40852554135449$ | Ostr- modif | | Nan | | Nan |
| Space containing the root: | Kou1 | | 4 | | 3 |
| $(1.8, +\infty)$ | Kou2 | | Nan | | Nan |
| | Kou3 | | 8 | | 6 |
| $f_{10}(V) = 100\,V^3 - 25.25394\,V^2$ $+1.36\,V - 0.00432888$ [22] | Newton | $[0.1656, 0.1856]$ | 7 | $[0.169, 0.1856]$ | 4 |
| | Ostrowski | | Nan | | 3 |
| Root: $V_n = 0.17714606941487$ | Ostr- modif | | 3 | | 2 |
| | Kou1 | | 3 | | 2 |
| Space containing the root: | Kou2 | | - | | Nan |
| $(1.14, +\infty)$ | Kou3 | | 3 | | 2 |

## 5. Concluding Remarks

In the article, we propose new interval modifications of some known iterative methods for solving nonlinear equations with fifth order of convergence. The proposed methods are compared with the known formulas of Newton and Ostrowski. Experiments show, that the methods we choose make a small number of iterations and are suitable for finding a real root of a non-linear equation within a selected interval.

**Author Contributions:** The contributions of all of the authors have been similar. All of them have worked together to develop the present manuscript.

**Funding:** This research received no external funding.

**Conflicts of Interest:** The authors declare no conflict of interest.

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
