# Peer review of "Interval Methods with Fifth Order of Convergence for Solving Nonlinear Scalar Equations"

_axioms, doi:10.3390/axioms8010015_

Round 1

Reviewer 1 Report

The topic of the article is very interesting. The article is mainly well written and the results are clear to be published in this well-known Journal.

Anyway, there are in the structure of the article some problems, which the authors have to solve:

·      First of all, the second point of Theorem 4 is not correct in its present form, so it must be revised and completed.

·      Then, a real-world application should be added in terms of show the applicability of the method not just in academic examples.

·      Is it posible to extend the method to systems of equations?

·      In the numerical examples, tolerance is not correct 

·      Moreover, the whole paper should be revised in order to correct tipos and missprints.

·      What about the dynamical behavior of the method?

·      And related to the efficiency of the method?

·      In equation (20) the symbol of derivate is not superscript. In general, the syntaxis must be revised. Small errors are seen.

·      Finally, references are too old, there exist lot of references related to higher-order iterative methods such us:

R Behl, Í Sarría, RG Crespo, ÁA Magreñán, Highly efficient family of iterative methods for solving nonlinear models, Journal of Computational and Applied Mathematics 346, 110-132, 2019

IK Argyros, S George, Ball convergence for Steffensen-type fourth-order methods, International Journal of Interactive Multimedia and Artificial Intelligence 3 (4), 27-42, 2015.

IK Argyros, D González, Local convergence for an improved Jarratt-type method in Banach space, International Journal of Interactive Multimedia and Artificial Intelligence 3 (4), 20-25, 2015

Author Response

First of all we want to thank to the reviewer for the detailed analysis of the developments in this paper as well as for the evaluation of the significance of the original results inclosed in the paper.

Below are our answers to your suggestions.

Anyway, there are in the structure of the article some problems, which the authors have to solve:

·      First of all, the second point of Theorem 4 is not correct in its present form, so it must be revised and completed.

Answer: Thank you for your careful reading which help us to obtain an improved version of the paper. Corrected.

·      Then, a real-world application should be added in terms of show the applicability of the method not just in academic examples.

Answer: Thank you for the remark. The introduced iterative interval schemes are applicable for the real-world applications such as an example for Van der Waals’ equation:

http://assets.openstudy.com/updates/attachments/57fd12d6e4b05a233fe6eb2c-cuanchi-1476213422408-van_der_waals1.pdf

In the paper we add two variants of Van der Waals’ equation presented through $f_9, f_{10}$ functions from Table 1.

·      Is it possible to extend the method to systems of equations?

Answer:  We agree your suggestion that the developments in this paper should be extended to the systems of equations.It will be a subject to the future work.

·      In the numerical examples, tolerance is not correct 

Answer: We rewrite the text “We use the stop criterion $rad(X)=\frac{1}{2}.(b-a)<tol$“ in the form “We use the stop criterion $wid(x^{(k)}) < tol$ for all interval methods.”  

·      Moreover, the whole paper should be revised in order to correct tipos and missprints.

Answer:  It is done. Thanks.

·      What about the dynamical behavior of the method?

Answer:  We have no investigation for the dynamical behavior of the proposed methods. It is our plan for the future work.

·      And related to the efficiency of the method?

Answer:  We introduce the efficiency index for Kou methods (4) and (12).

·      In equation (20) the symbol of derivate is not superscript. In general, the syntaxis must be revised. Small errors are seen.
Answer: Thank you very much for a comment. The superscript is a comma. There is an unnecessary right bracket. Corrected! 

·      Finally, references are too old, there exist lot of references related to higher-order iterative methods such us:

R Behl, Í Sarría, RG Crespo, ÁA Magreñán, Highly efficient family of iterative methods for solving nonlinear models, Journal of Computational and Applied Mathematics 346, 110-132, 2019

IK Argyros, S George, Ball convergence for Steffensen-type fourth-order methods, International Journal of Interactive Multimedia and Artificial Intelligence 3 (4), 27-42, 2015.

IK Argyros, D González, Local convergence for an improved Jarratt-type method in Banach space, International Journal of Interactive Multimedia and Artificial Intelligence 3 (4), 20-25, 2015

Answer:  According to your comments we have introduced The suggested papers of the Reference list.

Reviewer 2 Report

Some interval methods are presented to of fifth order to solve equations on R.

It is an interesting paper.

Author Response

Answer: Thank you for the careful reading of the paper and for the evaluation of the relevance of the obtained

results.

Reviewer 3 Report

The paper is well written and so interesting. I have carefully checked all calculations working entirely so well. Therefore I recommend its publication in this form 

Author Response

Answer: Thank you the reviewer for his favorable appreciations on the paper

Round 2

Reviewer 1 Report

Accept as is